**communications**

**biology**

# Structural and functional insights into δ-poly-L-ornithine polymer biosynthesis from *Acinetobacter baumannii*

Ketan D. Patel [1] & Andrew M. Gulick [1✉]

Cationic homo-polyamino acid (CHPA) peptides containing isopeptide bonds of diamino acids have been identified from Actinomycetes strains. However, none has been reported from other bacteria. Here, we report a δ-poly-L-ornithine synthetase from *Acinetobacter baumannii*, which we name PosA. Surprisingly, structural analysis of the adenylation domain and biochemical assay shows L-ornithine as the substrate for PosA. The product from the enzymatic reaction was purified and identified as poly-L-ornithine composed of 7-12 amino acid units. Chemical labeling of the polymer confirmed the isopeptide linkage of δ-poly-L-ornithine. We examine the biological activity of chemically synthesized 12-mer δ-poly-L-ornithine, illustrating that the polymer may act as an anti-fungal agent. Structures of the isolated adenylation domain from PosA are presented with several diamino acids and biochemical assays identify important substrate binding residues. Structurally-guided genome-mining led to the identification of homologs with different substrate binding residues that could activate additional substrates. A homolog from *Bdellovibrionales* sp. shows modest activity with L-arginine but not with any diamino acids observed to be substrates for previously examined CHPA synthetases. Our study indicates the possibility that additional CHPAs may be produced by various microbes, supporting the further exploration of uncharacterized natural products.

[1] Department of Structural Biology, Jacobs School of Medicine and Biomedical Sciences, University at Buffalo, SUNY, Buffalo, NY 14203, USA.
✉email: amgulick@buffalo.edu

Nonribosomal peptide synthetase (NRPS) enzymes, usually composed of condensation, adenylation, and thiolation domains, are involved in peptide natural product biosynthesis to produce antibiotics and siderophores[1]. Several variations to canonical NRPS modules exist which are commonly referred to as "NRPS-like" proteins[2]. These modules contain an adenylation domain, a carrier domain, and a C-terminal thioesterase or reductase domain, or more rarely a condensation domain[3,4]. ε-Poly-lysine synthetase (εPLs) and its homologs are similar NRPS-like modules that have a terminal membrane-bound domain instead of a thioesterase domain[5]. Some *Streptomyces* species employ an εPLs to produce polymers of cationic amino acids that are joined via isopeptide bonds between the carboxylate and the side chain amine[6]. These polymers can reach 20–35 residues in length. These polymers contain antimicrobial activity that is dependent on length, as a series of poly-ε-Lys shows a dramatic change in MIC against *E. coli* at a length of 9 lysine residues[7]. While conventional peptide poly-α-Lys also showed antibacterial activity, the activity of the polymers joined via isopeptide bonds was consistently more active against a variety of Gram-negative bacteria. Building on these early studies with εPLs, homologous enzymes that produce polymers of 2,3-diaminopropionate (DAP)[8] and both D- and L-isomers of 2,4-diaminobutyrate (DAB)[9] have been reported. A recent addition to the cationic homopolymeric amino acids (CHPA) family aided by the discovery of a gene encoding a lysine-2,3-aminomutase adjacent to a εPLs homolog has further extended the family of natural CPHAs to include ε-poly-β-lysine[10]. Finally, an unusual branched polymer, poly[β-(L-diaminopropionyl-L-diaminopropionic acid)] has also been identified[11].

Early studies of the biosynthesis of poly-ε-Lys showed that inhibitors of translation did not affect biosynthesis and that activation involved adenylation of the lysine building blocks, suggesting a potential involvement of an NRPS-like pathway[12]. Subsequently, Hamano and colleagues characterized εPLs, demonstrating that the cationic amino acid polymers are produced via a remarkable family of three-domain NRPS-like proteins[5]. The ~730 residue C-terminal domain had not previously been observed in NRPS proteins and failed to show significant homology to any known enzyme. Heterologous expression of the adenylation and carrier domains alone failed to produce ε-poly-Lys, demonstrating the involvement of this C-terminal domain in the polymerization reaction. Secondary structure predictions suggested that the C-terminal domain contained six transmembrane helices that were organized in three pairs. The three intervening sequences, which were ~180 residues in length and exhibited weak sequence homology, were proposed to form three "tandem soluble domains"[5]. The structure of the N-terminal adenylation domain of εPLs[13] illustrates that the protein adopted the conventional fold observed for NRPS adenylation domains and identifies substrate binding residues.

While natural CHPAs built from diamino acids with side chain lengths of four (lysine), two (2,4-diaminobutyrate), and one (2,3-diaminopropionate) methylene units linking the main chain Cα to the side chain amine have been reported, polymers built from ornithine (Orn), the diamino acid containing a primary amine side chain harboring three methylene units, have to date not been observed. Here we report our investigation of an εPLs homolog from the human pathogen *Acinetobacter baumannii* which produces δ-poly-L-ornithine. We have cloned and expressed the full-length protein and characterized biochemical activity and the structure of the adenylation domain. Activity analyses show that the *A. baumannii* homolog activates L-Orn and the full-length protein produces isopeptide polymers in vitro. Sequence analysis of the amino acid binding pocket of the adenylation domain guided mutagenesis, illustrating key residues that are involved in the determination of substrate specificity. We demonstrate that δ-poly-L-ornithine exhibits potent antifungal activity. Finally, we present bioinformatic analysis of this enzyme family, suggesting that additional amino acid polymers may be produced by as yet uncharacterized members of this family.

## Results

**Identification of substrate for the εPLs homolog from *A. baumannii*.** Considering the cationic nature of the polymers produced by εPLs homologs, the protein family is referred to as cationic homopolymeric amino acids synthetases (CHPAs). While most of the prior studies on CHPA synthetases have focused on genes from Actinomyces, species that are known to be prolific natural product producers, we chose an εPLs homolog from the Gram-negative human pathogen *Acinetobacter baumannii*. *A. baumannnii* contains a single εPLs gene that is present as a single gene operon, with no obvious neighboring, cotranscribed genes (Supplementary Fig. 1). We cloned the gene from *A. baumannii* strain AB307[14], a clinical isolate that we have previously used for studies of other NRPS proteins. PKS/NRPS analysis (http://nrps.igs.umaryland.edu/)[15] or Anti-SMASH 3.0[16,17] could not predict the substrate for adenylation domain of the *A. baumannii* εPLs homolog. Analysis of the Stachelhaus code[18] residues identified the lysine-activating adenylation domain from bacitracin synthetase module 1 as the closest homolog. The N-terminal adenylation domain of the *A. baumannii* homolog (residues 1–512) was cloned, expressed in *E. coli* BL21 (DE3), and purified to homogeneity for enzymatic assay and crystallization. The potential preference of a cationic amino acid substrate was supported by a crystal structure, described below (Fig. 1a), which confirmed the presence of negatively charged Glu221 and polar residues Thr304 and Ser313 at the side chain binding pocket (Fig. 1b). This supported an amino acid with positively charged group in the side chain as the probable substrate.

Enzyme activity assay using coupled NADH consumption[19] was performed with the truncated adenylation domain and standard proteinogenic and non-proteinogenic amino acids (Supplementary Figs. 2 & 3) as well as L-ornithine and a diastereomeric mixture of L/D-2,4-diaminobutyric acid (DAB) (Fig. 1c, d). The adenylation domain showed modest activity with L-lysine and a reproducible low level of activity with L-arginine and L-histidine. The enzyme assay with L-ornithine showed the highest activity while low activity was observed with DAB (Fig. 1c). This suggested L-ornithine was the most preferred substrate for the εPLs homolog from *A. baumannii*. Michaelis-Menton kinetics showed a preference in the apparent $k_{cat}/K_M$ of over 160-fold for L-ornithine compared with L-lysine (Table 1). The apparent $k_{cat}/K_M$ for D-ornithine was 250 $M^{-1} s^{-1}$, indicating a modest preference for the L-configuration by the *A. baumannii* adenylation domain.

**The *A. baumannii* homolog produces δ-poly-L-ornithine.** The C-terminal domain of εPL synthetase (εPLs) is proposed to be membrane-bound. Sequence alignment of NRPS terminal domain from *A. baumannii* homolog with the εPLs protein from *Streptomyces albulus* showed 30% similarity and 20% identity, with similar transmembrane helices that are separated by three segments of ~180 residues, that were described previously as tandem condensation domains[5]. The *S. albulus* εPL synthetase adenylation domain has similar Stachelhaus code residues to that from *A. baumannii* with alanine at the place of Met218. Our biochemical analysis on the adenylation domain in isolation prompted us to test for the ability of the full-length *A. baumannii* enzyme to produce poly-L-ornithine. Full-length εPLs homolog protein

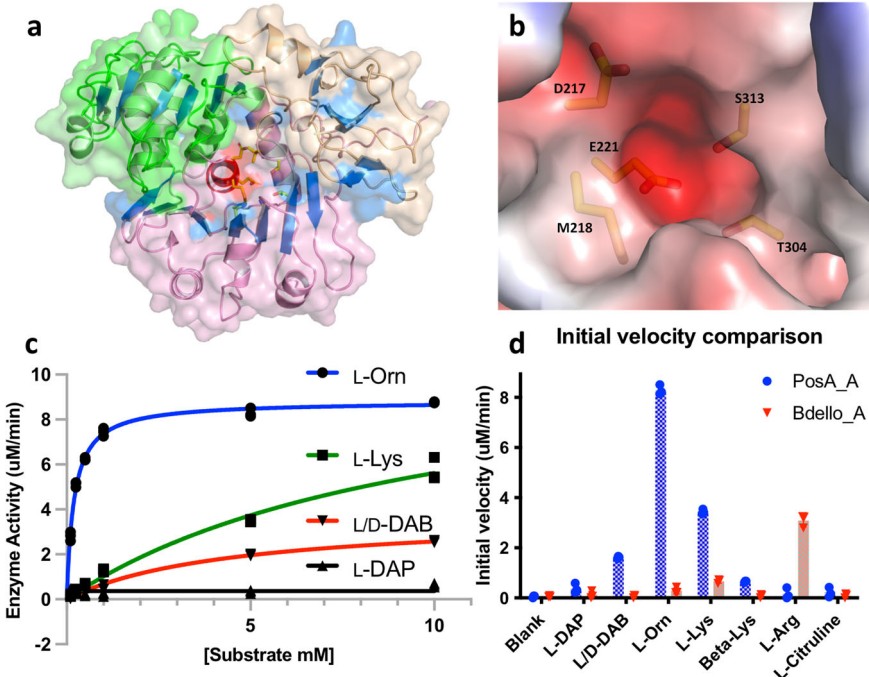

**Fig. 1 Structure and functional characterization of adenylation domain from PosA. a** PosA adenylation domain crystal structure showing N-terminal core domain of PosA adenylation domain with central helix colored red and side chains depicted for substrate binding residues. **b** Substrate binding pocket of PosA adenylation domain showing negatively charged and polar residues indicating positively charged amino acid as probable substrate. **c** Comparison of Michaelis-Menten plots for four diamino positively charged substrates indicating L-ornithine as the most preferred substrate and (**d**) Initial velocity comparison for adenylation domains of Pls homologs from Acinetobacter baumannii (PosA) and Bdellovibrionales sp. (Bdello_A). PosA showed the highest activity with L-ornithine and Bdello_A with L-arginine. Triplicate points are shown in panels c and d.

**Table 1 Apparent steady state kinetics of wild type and mutant PosA with different substrates[a].**

| Protein | Substrate | Protein conc. (µM) | $V_{max}$ (µM/min) | $K_M$ (mM) | $k_{cat}$ (sec$^{-1}$) | $k_{cat}/K_M$ (M$^{-1}$sec$^{-1}$) |
|---|---|---|---|---|---|---|
| WT | L-Orn | 1 | 7.74 ± 0.13 | 0.24 ± 0.02 | $(1.29 ± 0.03) \times 10^{-1}$ | $5.38 \times 10^2$ |
| WT | D-Orn | 1 | 6.76 ± 0.09 | 0.45 ± 0.02 | $(1.13 ± 0.02) \times 10^{-1}$ | $2.50 \times 10^2$ |
| WT | L-Lys | 5 | 8.46 ± 0.25 | 8.59 ± 0.68 | $(2.82 ± 0.08) \times 10^{-2}$ | $3.28 \times 10^0$ |
| T304A | L-Orn | 5 | 10.4 ± 0.3 | 2.12 ± 0.26 | $(3.47 ± 0.10) \times 10^{-2}$ | $1.64 \times 10^1$ |
| S313A | L-Orn | 8 | 45.7 ± 1.6 | 32.2 ± 3.1 | $(9.52 ± 0.33) \times 10^{-2}$ | $2.96 \times 10^0$ |
| E221A | L-Orn | 20 | --[b] | -- | -- | -- |
| D217A | L-Orn | 20 | --[b] | -- | -- | -- |
| M218A | L-Orn | 1 | 14.5 ± 0.2 | 0.51 ± 0.02 | $(2.42 ± 0.03) \times 10^{-1}$ | $4.74 \times 10^2$ |
| M218A | L-Lys | 1 | 7.06 ± 0.32 | 4.26 ± 0.54 | $(1.18 ± 0.05) \times 10^{-1}$ | $2.76 \times 10^1$ |

[a]Values are presented as standard errors from triplicate measurements of six to ten substrate concentrations.
[b]No detectable activity.

from *A. baumannii* was heterologously expressed in *E. coli* BL21 (DE3). Purification with detergent NP40, as used for εPLs[5], yielded soluble protein, while purification without detergent resulted in aggregation (Supplementary Figs. 4 & 5). This supported the hypothesis that the terminal domain of εPLs homolog is a membrane-embedded domain.

An enzymatic assay using full-length protein, adapted from the protocol of Yamanaka et al. [5], was performed and subjected to LC/MS analysis. Purification of the enzyme PosA and the enzymatic assay were performed using n-dodecyl-β-D-maltopyranoside (DDM) detergent. To purify and concentrate poly-L-ornithine from the enzymatic mixture, the cationic nature of the polymers was leveraged through purification over an SP cation exchange column. Absorbance peak at 210 nm, corresponding to peptide, was observed in 1 M NaCl elution only in the L-ornithine reaction (Fig. 2a). Fractions from this peak were subjected to LC/MS analysis, revealing a peak that comigrated with a commercial

standard of a 12-mer of δ-poly-L-ornithine (Fig. 2b, Supplementary Fig. 6), demonstrating that the homolog from *A. baumannii* functioned as poly-L-ornithine synthetase. Since L-lysine showed some activity in the adenylation assays, an enzymatic reaction with PosA protein and L-lysine was performed. The assays showed no peptide peak on SP cation exchange chromatography (Fig. 2a) nor the presence of lysine polymers with LC/MS (Fig. 2b, d, Supplementary Figs. 7, 8, and Supplementary Table 1). An optimized isocratic separation (Supplemental Fig. 9) confirmed the presence of polymers of ornithine ranging from 7 to 12 units in length (Fig. 2c).

The precedent established by the ε-poly-L-Lys and ε-poly-L-β-Lys synthetases, among others, suggested that the ornithine polymers were likely joined via isopeptide bonds between the δ amine of one ornithine and the carboxylate of the next. To examine this, we used chemical labeling with dansyl chloride to label the free amines of the peptide followed by hydrolysis and

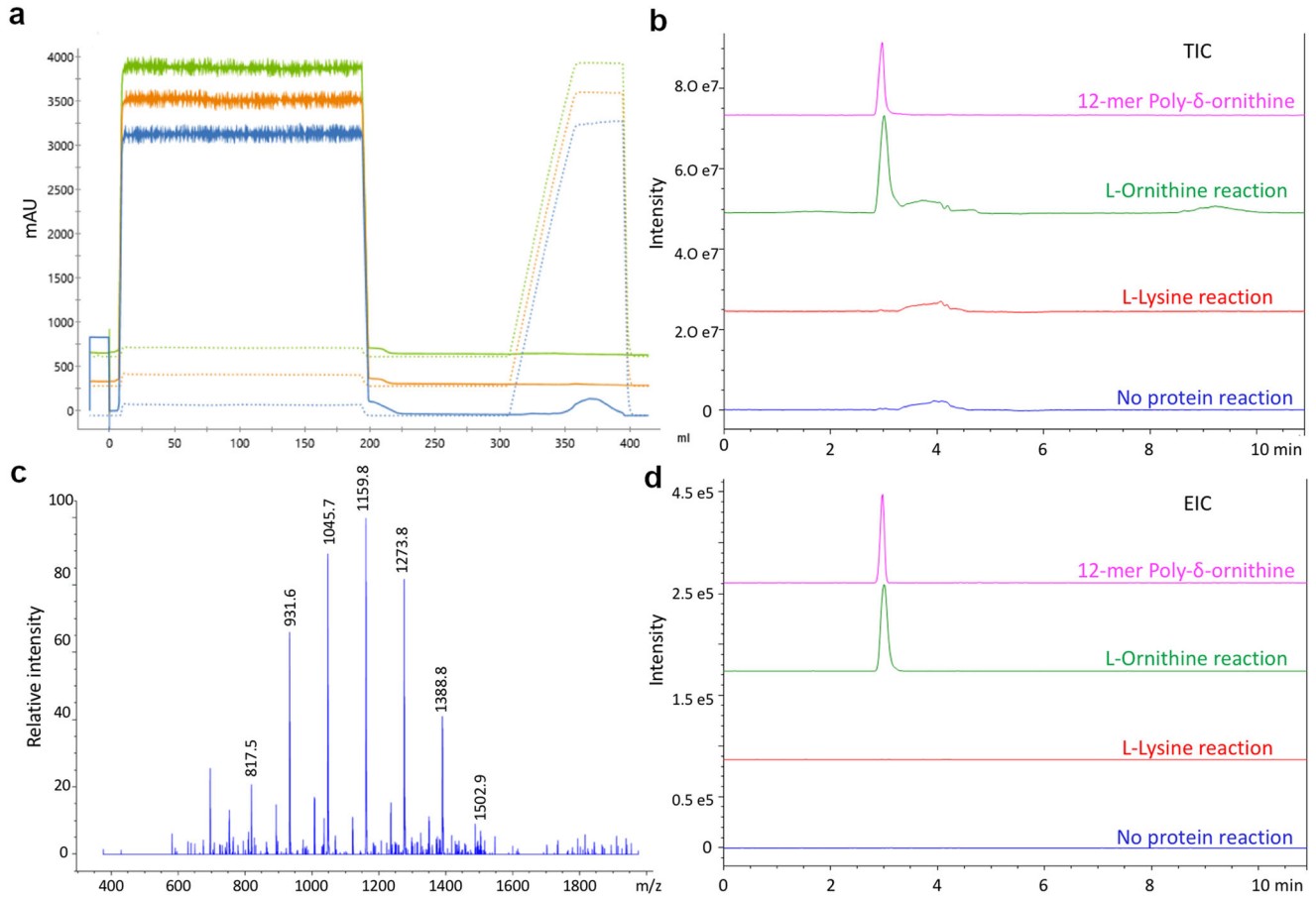

**Fig. 2 Purification and identification of poly-L-ornithine from PosA enzymatic reaction. a** SP cation-exchange chromatogram overlay showing purification of δ-poly-L-ornithine from PosA enzymatic reaction. Absorbance at 210 nm was used to check for the presence of peptide polymer. A peak for 210 nm was observed in PosA reaction with L-ornithine (blue) in 1 M NaCl elution, but not in PosA reaction L-lysine reaction (green) or control reaction lacking protein (orange). **b** Fractions from SP cation-exchange chromatogram 1 M NaCl elution were examined by LC/MS. TIC chromatograms from protein-free control, L-lysine reaction, and L-ornithine overlaid with 12-mer δ-poly-L-ornithine standard runs. **c** MS spectra from the peak of L-ornithine reaction showing 817.7, 931.6, 1045.7, 1159.8, 1273.8, and 1388.8 [M + H] masses representing 7-mer, 8-mer, 9-mer, 10-mer, 11-mer and 12-mer of δ-poly-L-ornithine respectively. MS Spectrum from 6.3 min peak of a large-scale separation and slower flow rate (Supplementary Fig. 9) LC/MS separation. **d** EIC chromatograms for 1386:1388 m/z masses representing 12-mer of poly-L-ornithine in commercial standard run (magenta) and L-ornithine reaction (green), but not in L-Lysine reaction (red) or protein-free control reaction (blue).

LC/MS detection of the labeled monomers. Because of the availability of the free amines, the α-poly-L-ornithine peptide produces δ-dansyl ornithine while the δ-poly-L-ornithine peptide produces α-dansyl ornithine (Supplementary Fig. 10, Supplementary Table 1). Dansylation of free ornithine with a 1:1 molar ratio of label:ornithine resulted in two peaks of singly labeled dansyl ornithine (Fig. 3a, b, Supplementary Fig. 11), as well as a peak of dansyl hydroxide (Fig. 3a, Supplementary Figs. 11, 12). The two peaks (12.6 and 13.5 min) of labeled ornithine showed the correct mass (Supplementary Fig. 13) and represent α- or δ-labeled ornithine. The identity of the two peaks was determined by dansylation and hydrolysis of commercial standards of 12-mers of poly-α-ornithine and poly-δ-ornithine, which each exclusively produced one of the two peaks (Fig. 3a, b).

With the chemical nature of the two dansyl ornithine peaks identified, the poly-L-ornithine product of the PosA reaction was labeled with dansyl chloride and hydrolyzed. The resulting LC-MS trace showed the presence of α-dansyl ornithine with no detectable δ-dansyl ornithine (Figs. 3a, 3b). Peaks with mass consistent with doubly and triply labeled ornithine dimers and trimers due to incomplete hydrolysis were also observed (Fig. 3 and Supplementary Fig. 13). Thus, *A. baumannii* protein can be described as a δ-poly-L-ornithine synthetase (δPOs). Adopting

the convention of other members of this family, the εPLs homolog from *A. baumannii* is referred to as δ-poly-L-ornithine synthetase A (PosA) protein that is encoded by the *posA* gene.

**Structures of PosA adenylation domain with substrates**. To better understand the substrate binding in PosA, we solved four structures of the isolated adenylation domain. We first solved the unliganded structure (Fig. 1a) using the phenylalanine activating domain of gramicidin synthetase PheA (PDB **1AMU**)[20] as a molecular replacement model. This refined, unliganded structure (PDB **8G95**) was used as a search model to solve three co-crystal structures with L-Orn, D-Orn, and L-Lys to identify substrate binding residues (PDB **8G96, 8G97**, and **8G98 respectively**). Data were collected to 2.2–2.5 Å for all four structures and refined to completion through iterative cycles of model building and refinement (Table 2). All four structures were solved in the same space group $P2_12_12_1$ with two protomers per asymmetric unit. Density for both ornithine isomers was unambiguous (Supplementary Fig. 14); the electron density for the lysine ligand was less complete for the carboxylate and amine, suggesting the longer substrate may adopt multiple conformations in the active site of the crystallized protein. As seen previously for other members of

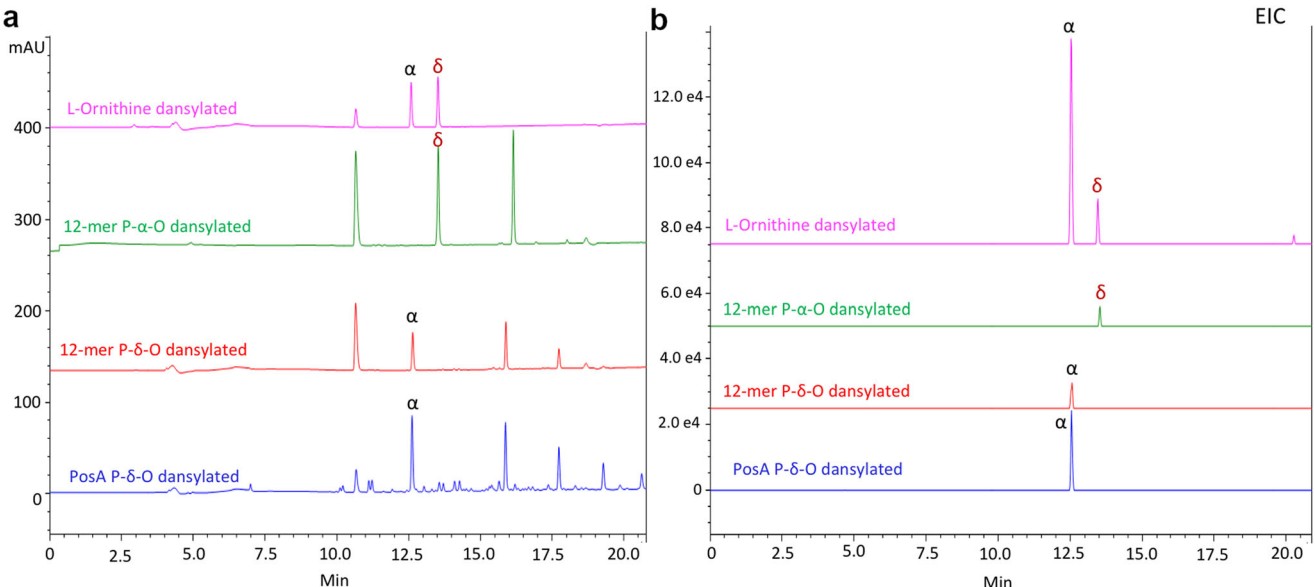

**Fig. 3 Identification of isopeptide-bonding in poly-L-ornithine from PosA using dansylation reaction. a** 254 nm absorbance chromatograms and (**b**) EIC (365:367) chromatograms for dansylation and hydrolysis reactions of free L-ornithine (magenta), 12-mer α-poly-L-ornithine standard (green), 12-mer δ-poly-L-ornithine standard (red), and enzymatically biosynthesized poly-L-ornithine from the PosA reaction (blue). A 254 nm absorbance peak, reflecting the presence of the dansyl moiety, at 10.6 min present in all reactions showed m/z mass of 252.0–253.9 matching to dansyl-OH [M + H]$^+$ 252.0 (Supplementary Figs. 12, 13). Absorbance peaks at 12.6 min and 13.5 min in L-ornithine dansylation reaction showed m/z mass of 366.1 matching to N-dansyl-L-ornithine (Supplementary Fig. 13). Dansylated and hydrolyzed δ-poly-L-ornithine standard (12-δPO) showed a peak at 12.6, while dansylated and hydrolyzed α-poly-L-ornithine standard (12-αPO) showed a peak at 13.5 min. Dansylated and hydrolyzed product from PosA reaction showed a peak at 12.6 min indicating that PosA produces δ-poly-L-ornithine.

this family of adenylate forming enzymes, including several firefly luciferases[21,22] and the free-standing adenylation domains BasE[23] and VinN[24], only the N-terminal or A$_{core}$ domain of the adenylation domain was observed in the crystal structures; no density was observed for the C-terminal A$_{sub}$ domain.

The structure of adenylation domain N-terminal subdomain (A$_{core}$) shows the typical acyl-CoA synthetase fold with three subdomains (Fig. 1a). Comparison with the recently solved[13] εPL synthetase adenylation domain (PDB **7WEW**), with which PosA shares 36% sequence identity over the A$_{core}$ subdomain, shows a root-mean-square displacement of 1.8 Å over 389 residues (Supplementary Fig. 15). The PosA sequence contains a 16-residue insertion at residue Tyr132 to Glu147 compared to εPL synthetase that results in an extra α-helix and an antiparallel β-strand, located over 20 Å from the active site. Substrates bind at the center of the core adenylation domain where residues mainly from subdomain I mediate binding (Fig. 4). The α-amino groups of all three substrates, L-ornithine, D-ornithine, and L-lysine, interact with of Asp217 in co-crystal structures (Fig. 4). This aspartate residue is conserved in all α-amino acid activating adenylation domains. Glu221 showed charged interaction with terminal δ-amine of ornithine substrates or the ε-amino group of lysine indicating an important role in binding. The polar side chains of Thr304 and Ser313 were also found to be placed at hydrogen bonding distances of 2.9 and 2.6 Å from the terminal amine group of L-ornithine.

**Mutational analysis of substrate binding residues.** Sequence analysis of the predicted Stachelhaus residues that form the substrate binding pocket was analyzed in the context of the experimental structure. Sequence analysis alone predicted a pocket consisting of "D M E H N G T V S K" for PosA. The structures with substrates revealed Asp217, Met218, Glu221, Thr304, and Ser313 as interacting residues with the diamine substrates. Mutagenesis of these residues was carried out to

evaluate their role in substrate binding. Mutation of individual residues showed Asp217 and Glu221 were the most important residues for substrate activation since no enzymatic activity was observed in these mutants even at high protein or substrate concentration (Table 1). Removal of the hydroxyl group from the side chains of Thr304 and Ser313 showed 33- and 183-fold reduction in catalytic efficiency in T304A and S313A mutants, respectively, indicating these residues are also important for binding (Table 1, Supplementary Fig. 16). Predicted residues His260, Asn280, and Val312 do not show interaction with D/L-Orn or L-Lys. As Met218 shows 3.9 Å hydrophobic interaction with C$_δ$ of L-ornithine, it may play an important role in the specificity of substrates, since CHPAs adenylation domains activating Lys have an alanine and PDAP synthetase has a phenylalanine residue at the respective position of Met218 in PosA (Supplementary Table 2).

Comparison of Stachelhaus code residues from CHPAs of various organisms revealed Asp217 and Glu221 as well conserved residues, while Met218, His260, and Asn280 are more variable residues. Many other CHPAs adenylation domains showed variation at Met218 position. Since His260 and Asn280 do not show interaction with substrates in δ-poly-L-ornithine synthetase adenylation domain, Met218 was considered for its role in substrate specificity. Mutation M218A showed minimal reduction in the apparent k$_{cat}$/K$_M$ for L-Orn. However, the apparent k$_{cat}$/K$_M$ increased nearly 10-fold for L-Lys for the M218A mutant (Table 1). This increase in efficiency and variability at this position in CHPAs family indicates Met218 is important for substrate specificity. Structural modeling of M218A mutation showed an increased volume of cavity in the substrate binding pocket supporting the role of methionine as a steric hindrance for substrates longer than ornithine (Supplementary Fig. 17). Overall, this indicates that residue at the Met218 position plays an important role for substrate specificity in PosA and other CHPA synthetase adenylation domains.

**Table 2 Data collection and refinement statistics for PosA adenylation domain structures.**

| Data Collection | Unliganded[a,b] | PosA, L-Orn[a,b] | PosA, D-Orn[a,b] | PosA, L-Lys[a,b] |
|---|---|---|---|---|
| PDB code | 8G95 | 8G96 | 8G97 | 8G98 |
| Beamline | SSRL 12-2 | GMCA, 23IDD | SSRL 12-2 | GMCA, 23IDD |
| Wavelength (Å) | 0.97946 | 1.03322 | 0.97946 | 1.03322 |
| Resolution range (Å) | 80.15–2.19 | 79.29–2.30 | 93.93–2.51 | 58.64–2.49 |
|  | (2.27–2.19) | (2.37–2.30) | (2.65–2.51) | (2.57–2.49) |
| Space group a, b, c (Å) | $P2_12_12_1$ | $P2_12_12_1$ | $P2_12_12_1$ | $P2_12_12_1$ |
|  | 64.36 | 63.97 | 63.83 | 63.48 |
|  | 94.10 | 92.78 | 93.92 | 91.97 |
|  | 152.94 | 152.64 | 152.25 | 152.24 |
| α, β, γ (°) | 90 90 90 | 90 90 90 | 90 90 90 | 90 90 90 |
| Total reflections | 609329 (61725) | 505942 (49183) | 209259 (20649) | 408887 (36625) |
| Unique reflections | 48260 (4762) | 39481 (4052) | 31440 (3078) | 31880 (3069) |
| Multiplicity | 12.6 (13.0) | 12.8 (12.1) | 6.7 (6.7) | 12.8 (11.9) |
| Completeness (%) | 99.83 (99.87) | 95.41 (99.95) | 98.26 (97.30) | 99.53 (97.99) |
| Mean I/sigma(I) | 12.58 (2.29) | 13.35 (0.89) | 15.04 (1.36) | 15.54 (1.27) |
| Rmerge | 0.1251 (1.322) | 0.1403 (3.238) | 0.0924 (1.505) | 0.1269 (2.241) |
| Rpim | 0.03693 (0.3802) | 0.04061 (0.9639) | 0.03841 (0.6179) | 0.0368 (0.6696) |
| $CC_{1/2}$ | 0.997 (0.86) | 0.999 (0.416) | 0.999 (0.75) | 0.999 (0.506) |
| Refinement |  |  |  |  |
| Resolution range (Å) | 50.18–2.19 | 76.32–2.30 | 48.91–2.51 | 58.64–2.49 |
| Reflections, refinement | 48220 (4757) | 39454 (4051) | 31373 (3064) | 31862 (3069) |
| Reflections, $R_{free}$ | 1998 (197) | 1998 (205) | 1994 (195) | 1999 (193) |
| $R_{work}$ | 0.2078 (0.3026) | 0.2072 (0.4166) | 0.1897 (0.3300) | 0.1919 (0.3107) |
| $R_{free}$ | 0.2258 (0.3803) | 0.2380 (0.4455) | 0.2391 (0.3762) | 0.2493 (0.3485) |
| Protein residues | 810 (2 chains) | 812 (2 chains) | 803 (2 chains) | 805 (2 chains) |
| Ligands of interest |  | 2 (L-Orn) | 2 (D-Orn) | 1 (L-Lys) |
| Other molecules | 4 EDO, 1 CAC | 2 glycerol | 12 glycerol | 12 glycerol |
| Water molecules | 201 | 124 | 144 | 103 |
| Rms (bonds) (Å) | 0.004 | 0.005 | 0.004 | 0.008 |
| Rms (angles) (°) | 0.99 | 1.12 | 0.99 | 1.27 |
| Rama. Favored (%) | 97.89 | 97.03 | 96.61 | 97.37 |
| Rama. Allowed (%) | 2.11 | 2.85 | 3.39 | 2.63 |
| Rama. Outliers (%) | 0.00 | 0.12 | 0.00 | 0.00 |
| Rotamer outliers (%) | 0.15 | 0.87 | 0.15 | 0.75 |
| Average B-factor (Å2) |  |  |  |  |
| overall, protein | 56.84 | 75.93 | 73.75 | 77.13 |

[a]Each structure was solved from a single crystal.
[b]Values in parentheses represent statistics for the highest resolution shell.

**Homo-polymer synthetase in other organisms**. BLAST and HMMER based searches for additional homologs showed the presence of CHPAs in a wide variety of bacterial phyla and several fungal genera, as well as a few species in cyanobacteria and archaea (Fig. 5, Supplementary Table 2). Surprisingly, some species from Metazoa, Alveota, and Amoebozoa also showed CHPAs homologous sequences. To find additional homopolymer synthetases, a protein sequence similarity network (SSN) was created using 548 sequences from diverse organisms (Supplementary Fig. 18). A large number of clusters indicated the diversity of sequences in homopolymer synthetases. Various bacterial classes from Actinobacteria and Proteobacteria phyla formed major clusters 1-3 and 5-8. Homopolymer synthetases from fungi formed major clusters 4 & 9, with sequence length <1500 residues. Additional sequences from Spirochetes, Archaea, and lower eukaryotes populated single node clusters. While homopolymer synthetases from many bacterial genera showed residues matching to εPL, δPO, or PDAP synthetases, SSN analysis further helped to identify unique signature sequences including several from Archaea, Rhizobia, Methylobacterium and other species with "D L E D I G T V T/V/I K" and in fungi *Botryobasidium botryosum* with "D I E S I G T V T K" as Stachelhaus code residues. Surprisingly cyanobacteria (clusters 45 & 52) also showed residues for εPL and PDAP synthesis. Although single node clusters were formed mostly from unique bacteria and lower eukaryotes showing known substrate binding residues, some homologs contained predicted binding residues that suggest they may activate additional substrates and hence may produce previously unreported polymers. For example, homologs from *Methylomarinum sp.* showed "D G E S I S V V N K" and *Bdellovibrionales bacterium* showed "D S E C Y G G V T K" active site pockets.

To explore the relationship among different CHPA synthetases, a phylogenetic tree was generated consisting of selected homologous sequences from all organisms. Sequences showing Stachelhaus code residues for εPLs and δPOs homologs made two separate branches, while the remaining sequences with PDAP/ PDAB (poly-DAP and poly-DAB) synthetases formed the largest group (Fig. 5). Several unique sequences with unknown substrate and εPLs homologs formed a small third group. The fourth group containing homologs of PDAPs, PDABs, εPBLs (ε-poly β-lysine synthetase), and unknown substrate activating CHPAs formed the largest group since they were found close to each other and difficult to distinguish on the basis of sequence alignment. εPBLs formed a separate branch in group 4 indicating they are unique among the group. Surprisingly, sequences showing different Stachelhaus code residues and probably activating an unknown substrate were found in three groups except group 2 for εPLs homologs. Most sequences from fungi formed a separate branch in the ornithine activating branch, while sequences from other

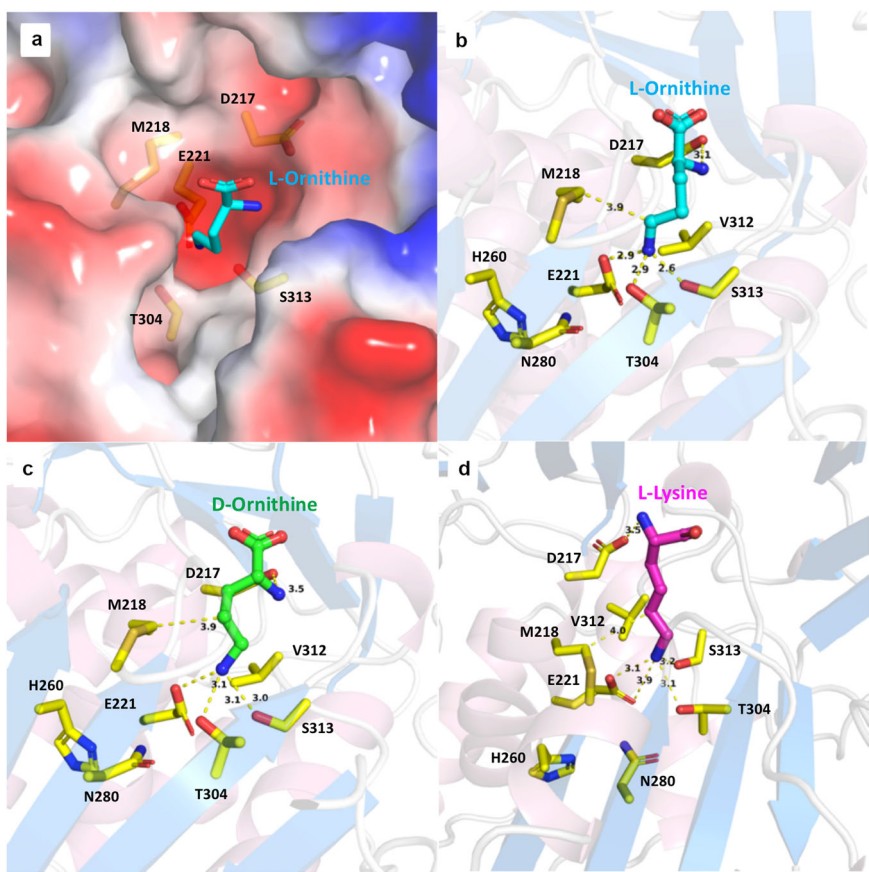

**Fig. 4 Active site structure of PosA adenylation domain bound to ligands. a** L-ornithine substrate binding in the negatively charged pocket of PosA adenylation domain. **b** L-ornithine substrate molecule interaction with substrate binding residues. Residues Asp217 and Glu221 make charged interaction with α-amine and δ-amine of the substrate L-ornithine respectively. Thr304 and Ser313 interact with the δ-amine while Met218 forms van der Waals interactions with δ-carbon of L-ornithine. Residues His260 and Asn280, although part of Stachelhaus code, do not show any interaction with the substrate. **c** D-Ornithine and (**d**) L-Lysine make similar interactions as L-ornithine with substrate binding residues.

eukaryotes and archaea with unknown substrates belonged to the fourth group. Overall, phylogenetic analysis indicates that several εPLs and δPOs homologs are distinct, while PDAPs, PDABs, εPBLs homologs are closer to each other and several CHPAs from three groups need to be investigated to find additional polymers.

**Homo-polymer synthetases with uncharacterized binding pockets may produce additional peptides.** Six homo-polymers of diamino substrates, including several that are based on propionate, as well as butyrate, and lysine are known in literature; δ-poly-L-ornithine is reported in the present investigation. However, sequence analysis showed alternate substrate binding pocket residues in the adenylation domains of uncharacterized enzymes (Supplementary Table 2) with a similar NRPS terminal domain. This suggested the activation of other substrates by these proteins and the potential biosynthesis of polymers that to our knowledge have not yet been reported. To experimentally verify whether variation in Stachelhaus code residues leads to the activation of diverse substrates, two additional proteins were examined. The CHPA protein sequence from *Bdellovibrionales bacterium* contains Stachelhaus code residues "D S E C Y G G V T K", which showed the presence of Ser at chain length determining Met218 position 2 and lacks supporting polar residues seen in PosA. A second CHPA protein from bacteria *Methylomarinum sp* contains Stachelhaus code residues "D G E S I S V V N K" that showed the presence of Gly at Met218 position, which might activate a larger substrate. A recently characterized εPLs homolog from *Streptoalloteichus hindustanus* NBRC 15115 with

Stachelhaus code residues as "D G E S I S V V N K" which matches that from *Methylomarinum sp* showed biosynthesis of an unusual polymer ε-Poly β-lysine synthetase[10]. Codon-optimized synthesized DNA for adenylation domains of the two genes were cloned and expressed in *E. coli* BL21 (DE3) strain. The adenylation domain from *Bdellovibrionales bacterium* (Bdello_A) expressed and purified well; however, that from *Methylomarinum sp* showed very low soluble protein. Purified protein from *Bdellovibrionales bacterium* was used for coupled NADH consumption assay to find the preferred substrate for activation. Bdello_A did not show higher activity with known four di-amino substrates but showed activation with L-arginine (Fig. 1d). This may indicate the preferred substrate could be a larger analog that would be chemically suitable for polymerization. Hence, our results indicate the potential that proteins with diverse substrate binding residues may produce additional polymers. Future investigations are underway to characterize these proteins and the antimicrobial activity of new polymers.

**Biological activities of δ-poly-L-ornithine.** Poly-lysine and PDAB found in *Streptomyces* strains exhibit antimicrobial properties. The presence of δ-poly-L-ornithine synthetase in a pathogenic strain of *A. baumannii* 307 prompted us to investigate the role of δ-poly-L-ornithine in various physiological functions. Since, a higher amount of δ-poly-L-ornithine could not be purified from the enzymatic reaction of PosA, chemically synthesized 12-mer δ-poly-L-ornithine (12-δPO) was used for the assays. Considering the ability of δ-poly-L-ornithine polymers in the

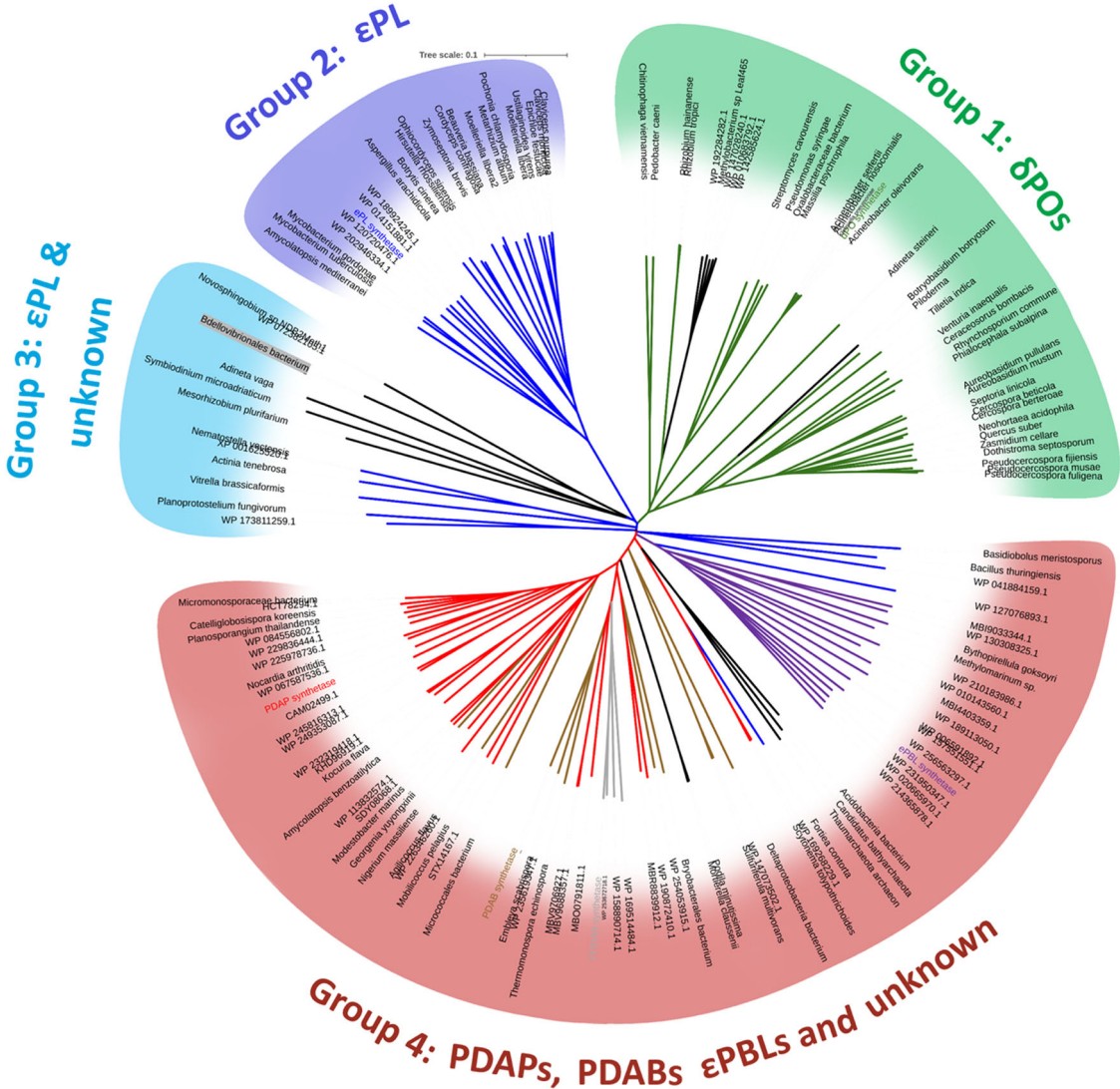

**Fig. 5 Unrooted phylogenetic tree of 148 CHPAs members from different bacteria, archaea, fungi, amoebozoa, metazoa, and Alveota.** Tree branches for sequences showing Stachelhaus code residues matching to PosA/ δPOs (δ-Poly-L-ornithine synthetase) are colored green (Group 1), those matching to εPLs (ε-Poly lysine synthetase) are in blue (Group 2 & 3), matching to Poly diamino-butyrate synthetase (PDAB) and Poly diamino-propionate synthetase (PDAP) are in red and brown and unknowns in black (Group 4). εPBLs (ε-Poly β-lysine synthetase) homologs formed a small separate group within group 4. Tree included all sequences from Supplementary Table 2 and others selected from BLAST and HMMER search using six known CHPAs as a query.

attachment of cells to the surface, the role of δ-poly-L-ornithine produced by *A. baumannii* in biofilm formation ability was examined. A knockout mutant of *posA* gene was generated with PCR mutagenesis to compare biofilm formation efficiency with the wild type. Growth of *A. baumannii* in static or shaken cultures was similar for both WT and mutant. The *posA* mutant showed slightly higher biofilm formation compared to WT (Supplementary Fig. 19).

Since polymers of α-amino bonded L-ornithine are used to improve the efficiency of substrate penetration in eukaryotic cells[25], the 12-mer δ-amino bonded L-ornithine (12-δPO) was examined in several additional tests. First, the addition of 12-δPO did not show toxicity to human cells compared to the control (Table 3). Motivated by the antimicrobial properties of εPLs and PDAB against bacteria and fungi, respectively, we tested the effect of ornithine isopeptide polymers on microbial growth. 12-δPO showed minimal activity against *Pseudomonas aeruginosa* and *Klebsiella pneumoniae* but showed growth inhibition against

**Table 3 Effect of δ-poly-L-ornithine on microorganisms and eukaryotic cell line.**

| Cell line | MIC (g/L) |
|---|---|
| Bacteria | |
| *Pseudomonas aeruginosa 01* | >0.30 |
| *Klebsiella pneumoniae* | >0.30 |
| Fungi | |
| *Saccharomyces cerevisiae* | 0.15 |
| *Cryptococcus neoformans* | 0.15 |
| *Candida albicans* | 0.30 |
| Eukaryotic cells line | |
| OVCAR-3 cell line | >0.50 |

MIC was calculated as no growth compared to control after two days.

*Saccharomyces cerevisiae* (Table 3). We further tested 12-δPO against pathogenic strains of fungi, which both were sensitive to 12-δPO, with *Cryptococcus neoformans* showing greater MIC than *Candida albicans*.

**Structural modeling of the uncharacterized C-terminal domain**. There is currently no structural information about the C-terminal domain of the homopolymer synthetase enzymes. As noted originally[5], the domain contains ~730 residues with six predicted transmembrane (TM) helices that are organized in three pairs that are expected to insert into the membrane. The amino acid sequences located between each pair of TM helices have been labeled the C1, C2, and C3 domains[5]. To further provide insight into the potential structure of the polymerization domains, we submitted the sequence of the PosA C-terminal domain to AlphaFold[26] and RoseTTAFold[27] algorithms to identify a potential fold of the C-terminal polymerization domain. Both programs returned nearly identical structures. The predicted structure (Supplementary Fig. 20) forms six bent helices that are suggested to be embedded in the membrane. The three intervening sequences, denoted previously as tandem C1, C2, and C3 condensation domains, pack together into a single large domain. This modeled domain contains approximate three-fold symmetry, reflecting the sequence similarity in the full polymerization domain. Atop the helical bundle are three left-handed β-helices of five turns each. The three β-helices pack against each other in a manner seen previously in several sugar nucleotide acyl transferases[28–31]. Between each pair of transmembrane α-helices and the subsequent β-helices are predicted to be two α-helices that lie in the core of the protein and a β-strand that forms a three stranded β-sheet with two additional strands that interrupt the second and third turns of the β-helix. While the complete structural characterization of the polymerization domain will require experimental support for this model, it suggests that the three regions that span the transmembrane helices may not be independent domains but rather form a single domain to play a catalytic role in the polymerization of the homopolymers of amino acids.

## Discussion

Of all the potential repeating polymers that could be formed from amino acids, only a few homo-polymers and homo-oligomers have been naturally identified[6]. The amide bond formation in these homo polymers is mediated by different enzymes. ADP-forming amide ligases produce various chain lengths of poly-α or γ-glutamic acid[32,33], whereas a stand-alone adenylation domain produces polymers of β-lysine used in streptothricin biosynthesis[34]. Additionally, the cyanophycin synthetase enzyme produces polymers of aspartic acid and arginine, in which polymers are formed between aspartic acid residues that have each been modified by an isopeptide bond to the main chain amine of an arginine residue[35,36]. ε-Poly-L-lysine (εPL), ε-Poly-L-β-lysine, γ-poly-L-diamino butyric acid, γ-poly-D-diamino butyric acid, β-poly-L-diamino-propionic acid (PDAP) and β-poly-L-diamino-propionyl-L-diamino-propionic acid are synthesized by NRPS-like protein having adenylation, carrier and a terminal transmembrane domain. In the present investigation, we report a δ-poly-L-ornithine synthesized by a homologous NRPS-like protein from *A. baumannii*. δ-poly-L-ornithine fills the carbon chain length gap in polyamine polymers from three-carbon in PDAP to six-carbon in εPL. This prompted us to compare the substrate binding pocket in all known CHPAs. This would allow us to understand the accommodation of various chain lengths in substrate binding pockets as well as substrate prediction.

We propose residue at Met218 position (2nd position in Stachelhaus code, Supplementary Table 2) plays an important role in determining the chain length of the substrate. PDAP/PDAB synthetase has a phenylalanine, PosA has a methionine, and PLs has an alanine, suggesting that a reduction in side chain size to accommodate the increase in substrate chain length from three carbons in PDAP to six-carbon in lysine (Supplementary Table 2). The mutation M218A in PosA led to increased activation for L-Lys, confirming the role of this position for substrate specificity in CHPAs. Glu221 in PosA is the most important to bond with the side chain amino group of substrates and is also conserved in all CHPAs. Residues such as Thr304 and Ser313 play a supporting role in binding the positively charged side chain of the substrate. Other Stachelhaus code residues His260 (position 4), Asn280 (position 5), and Val312 (position 8) were not found to directly interact with the substrate in PosA. This analysis indicates that CHPAs activating positively charged substrates have conserved positions for Asp217 and Glu221 (positions 1 and 3 of Stachelhaus code) while supporting residues may be present at any position. However, ornithine activating adenylation domains from other characterized NRPSs that are not involved in homopolymer synthesis like bacitracin synthetase module 2 (BacB-M2_A) shows the different position for Glu221 (position 4 of Stachelhaus code, His260 position in PosA), while that from coelichelin synthetase module 3 (CchH-M3_A) shows similar position to CHPAs (Supplementary Fig. 21). Therefore, it appears that while CHPAs may have conserved positions for Glu221, other adenylation domains may have the negative charged residue oriented from a different position to bind positive charged side chain of substrate indicating degeneracy of Stachelhaus code residues to bind same substrate.

The wide-spread presence of CHPAs proteins prompted us to investigate the role of homopolymers other than antimicrobial activity. α-Bonded poly-L-ornithine has been used as permeation enhancer for eukaryotic cells[37,38] or to improve attachment of cells on plastic or glass surfaces[39]. Long polymers of L-ornithine are also used to promote differentiation of stem cells[40]. Our results show the 12-mer of δ-poly-L-ornithine synthesized in PosA enzymatic reaction has anti-fungal activity indicating a potential role in survival benefit for *A. baumannii*. Variation in length of poly-L-lysine has a major effect on its antibacterial properties. Shorter chains have minimal activity while chains of 25 to 35-mer have optimal bacterial growth inhibition. Hence, it would be interesting to investigate the length of δ-poly-L-ornithine produced by *A. baumannii*, since longer chain may have a higher anti-fungal activity or maybe a different role altogether. Synthetic poly(L-ornithine) was also found to have biofilm-disrupting capacity[41]. Similarly, our results showed that an *A. baumannii* 307 knockout strain of *posA* gene had a modest increase in biofilm formation compared to the wild type indicating δ-poly-L-ornithine may have a role in the detachment of cells from biofilm. However, a detailed investigation is needed to confirm the role of δ-poly-L-ornithine in *A. baumannii* biofilm destabilization or cell detachment.

The search for anti-microbial peptides may aid the discovery of new therapeutics. Identifying natural product analogs has been found to be a successful strategy for discovering effective antimicrobials[41]. With the advent of genomic data, many homologous sequences of εPL synthetase could be found. Our results indicate that additional homopolymers may be found in these homologs. Structure, sequence, and SSN/EFI analysis helped us to find uncharacterized CHPAs in *Bdellovibrionales bacterium* and *Methylomarinum sp* which may synthesize other homopolymers. Identification of residues important for substrate selectivity further helped us to list all CHPAs that could produce new polymers (Supplementary Table 2), which may facilitate the

discovery of other polymers synthesized by CHPAs. Surprisingly, the CHPAs proteins could be found not only in a variety of bacteria, cyanobacteria, fungi, and archaea but also in some lower eukaryotic phyla like Amoebozoa, Metazoa, and Alveota. Hence, we propose that antimicrobial homopolymer synthesis is a widespread strategy used by prokaryotes and some lower eukaryotes to adapt to competitive environments.

## Methods

**Cloning and gene annotation**. Genomic DNA from *A. baumannii* 307-0294 strain was isolated using NEB Monarch kit as per manufacturer's instructions. PCR amplification of full length *posA* gene (GENBANK accession code: ATY43264.1, Supplementary Table 3), its adenylation domain (residues 1–512), and commercially synthesized DNA of *Bdellovibrionales bacterium* (OFZ39577.1) and *Methylomarinum sp.* (NOR71114.1) adenylation domain genes was done using NEB Phusion polymerase using primers as described in Supplementary Table 4. The sequences of codon optimized genes are included in Supplementary Table 5. Amplified products were cloned into modified pET15b vectors with N-terminal Poly-His5X tag and TEV cleavage site using InFusion cloning kit (Takara Bio). Ligation products were transformed directly into ultra-competent BL21 (DE3) cells and clones were selected using LB agar plates containing 100 ug/ml ampicillin. The PosA protein sequence is 99% identical to sequences from common laboratory strains of *A. baumanni* ATCC19606 and ATCC17978 (WP_001071468 and WP_001071474, respectively). In the ATCC17978 strain, the homolog is locus tag A1S_2651.

**Large scale protein purification**. For large scale protein expression, the primary culture was inoculated into six 2.8 L flasks each with 0.75 L LB medium, and induced by 0.5 mM IPTG at 0.7-0.8 OD600. After 16–18 h incubation at 20 °C and 250 rpm shaking, cells were harvested by centrifugation at 6000 rpm for 10 min. His-tagged proteins were purified over IMAC using lysis buffer (50 mM Tris-Cl, 400 mM NaCl, 0.5 mM TCEP, 20 mM imidazole, and 5% glycerol, pH 8.0) and elution buffer (50 mM Tris-Cl, 400 mM NaCl, 0.5 mM TCEP, 250 mM imidazole and 5% glycerol, pH 8.0). Cells were suspended in lysis buffer and lysed by sonication at 10 s ON and 15 s OFF cycle for 5 min ON time at 60% amplitude. Lysate was centrifuged at 40,000 rpm for 40 min at 4 °C and supernatant was filtered through 0.45 μ filter. The clear supernatant was loaded onto a single HisTrap FF 5 ml column for His-tag affinity purification. His-tagged proteins were eluted using 10 column volume of gradient with elution buffer. Eluted proteins were incubated with TEV protease (1:100 mg, TEV protease:His-tag protein) and dialyzed (50 mM Tris-Cl, 400 mM NaCl, 0.5 mM TCEP and 5% glycerol, pH 8.0) overnight at 4 °C. The following day, the cleaved protein suspension was cleared using 0.45 μ filter and passed over HisTrap FF 5 ml column for His-tag removal. Flow through was collected and concentrated using 15 ml Amicon centrifugal concentration tubes at 4000 rpm for 10 min. Once protein concentration reached 10–20 mg/ml, the protein sample was centrifuged at 14,000 rpm to remove aggregates, and the clear supernatant was loaded onto GF S200 XK 16/60 column for gel filtration equilibrated with a buffer containing 25 mM HEPES, 150 mM NaCl, 0.25 mM TCEP, pH 7.5. Fractions from the peak representing PosA adenylation domain were examined by SDS-PAGE. Fractions showing pure protein were pooled and concentrated up to 10 mg/ml for biochemical assay and 50 mg/ml for crystallization. Mutants S313A, T304A, and M218A were similarly purified using IMAC and gel filtration chromatography using the same buffers.

Full length PosA protein for the enzymatic assay was expressed and purified similarly using lysis buffer (50 mM Tris-Cl, 200 mM

NaCl, 0.2 mM TCEP, 20 mM imidazole 10% glycerol, 1% DDM (w/v), pH 8.0), equilibration buffer (50 mM Tris-Cl, 200 mM NaCl, 0.2 mM TCEP, 20 mM imidazole 10% glycerol, 0.2% DDM (w/v), pH 8.0), elution buffer (50 mM Tris-Cl, 200 mM NaCl, 0.2 mM TCEP, 250 mM imidazole 10% glycerol, 0.2% DDM (w/v), pH 8.0), dialysis buffer (50 mM Tris-Cl, 200 mM NaCl, 0.2 mM TCEP, 10 % glycerol, 0.2 % DDM (w/v), pH 8.0) and gel filtration buffer (50 mM Tris-Cl, 200 mM NaCl, 0.2 mM TCEP and 10% glycerol, 0.2% DDM (w/v), pH 8.0). DDM was purchased from Anatrace (Catalog no. D310). After lysis, IMAC purification, TEV-cleavage, and overnight dialysis, full-length PosA protein was treated with 100 nM Sfp phosphopantetheine transferase, 100 μM Coenzyme A, and 25 mM MgCl₂ for 2 h at 4 °C. Reaction mixture was run over IMAC column again to remove cleaved His-tag, tagged protein and Sfp protein followed by concentrating to 10 mg/ml and size exclusion chromatography over GF S200 XK 16/60 column for gel filtration.

**Crystallization and structure solution**. Truncated adenylation domain protein from PosA was buffer exchanged in 100 mM sodium cacodylate, 50 mM TAPS pH 8.5, 0.2 mM TCEP by gel filtration, and screened for crystallization. Long, rod-shaped crystals were obtained at 30 mg/ml with 0.1 M BisTris Propane pH 7.0, 0.1 M ammonium bromide, 20% PEG 20,000 at 20 °C. Further optimization to obtain well-diffracting crystals was done with 30–40% PEG 20,000, 0.1 M ammonium bromide, and 0.1 M BisTris Propane pH 7.0 in microbatch-under-oil plates. Co-crystallization with ligands was done by adding 2.5–5 mM of L-Orn, D-Orn, or L-Lys in the protein sample followed by setting crystallization drops. Crystals were cryo-protected by serial transfer into 8, 16, and 24% of glycerol in mother liquor and flash-frozen in liquid nitrogen. Remote data collection from single crystals was done at SSRL and APS. Data from unliganded adenylation domain crystal were processed to 2.1 Å resolution in the $P2_12_12_1$ space group (Table 2). Structure solution was performed by molecular replacement using Phaser in Phenix module and PheA (PDB ID: **1AMU**) as a search model. A solution with two protomers per asymmetric unit was obtained; however, density for C-terminal region of adenylation was not observed. Coot[42] and Phenix.refine[43] were used iteratively for manual rebuilding and refinement of the model respectively. Similarly, structure solution for substrate-bound adenylation domain was performed using unliganded structure as a search model. All liganded structures showed similar two protomers per asymmetric unit. The figures were prepared using PyMOL. The structure factors and atomic coordinates of the PosA adenylation domain are available at the Protein Data Bank (**8G95**, unliganded PosA, PDB; **8G96**, PosA bound to L-ornithine; **8G97**, PosA bound to D-ornithine; and **8G98**, PosA bound to L-lysine).

**Enzymatic reaction of PosA and purification of δ-poly-L-ornithine**. Enzymatic reaction for δ-poly-L-ornithine biosynthesis was first performed using full length PosA as described in Yamanaka et al. [5]. Since a low yield of δ-poly-L-ornithine was observed, PosA protein concentration was increased to 100 μg/ml. For detection of δ-poly-L-ornithine in LC/MS, 200 ml of enzymatic reaction was set up containing 100 mM TAPS-NaOH buffer, pH 8.5, 2 mM L-Orn (or L-Lys), 5 mM MgCl₂, 5 mM ATP, 1 mM DTT, 20% glycerol (v/v), 0.2% DDM (w/v), and 100 μg/ml PosA protein and incubated at 25 °C for 16 h. A negative control was set up without protein. After incubation, the pH was adjusted to 7.5 with 0.5 N HCl, and heat treatment was given at 100 °C in a water bath for 5 min to quench the reaction. Precipitants were removed by centrifugation at 40,000 rpm for 10 min at 4 °C followed by filtration with 0.45 μm filter. The filtrate was loaded

onto a single 5 ml SP cation exchange column equilibrated with buffer 20 mM TAPS pH 7.5 and 25 mM NaCl. Elution of δ-poly-L-ornithine was done using a linear gradient up to 1 M NaCl followed by 3 CV wash. Fractions corresponding to poly-L-ornithine peak in SP cation exchange chromatography from 1 M NaCl elution were further concentrated by drying 2.0 ml (aliquots of 250 ul) in the dry bath at 60 °C overnight. After complete drying, the sample was dissolved in 200 µl of 70% alcohol by pipetting and vortexing. The sample was again kept in the dry bath to evaporate alcohol until 50 µl of the solution was left followed by centrifugation at 14,000 rpm for 10 min and loading onto LC/MS. A Poroshell 120-EC C18 reverse-phase 4.6 × 100 mm column (Agilent) was used for LC/MS (Agilent 1260 Infinity II connected to MS module Agilent InfinityLab LC/MSD). HPLC method used an isocratic run of 50% acetonitrile in water with 0.1% formic acid for 10 min at 0.5 ml/min (Fig. 2) or 0.25 ml/min (Supplementary Fig. 9). Mass spectra analysis was done in positive-ion mode.

**Chemical characterization of poly-L-ornithine**. Chemical modification with dansyl chloride was adapted from Walker JM[44]. All dansylation reactions were performed in 2.0 ml HPLC amber glass vials. 20 µl of 1.66 mM 12-mer α-poly-L-ornithine standard or 12-mer δ-poly-L-ornithine or 20 ul of 10 mM L-ornithine was also used as standard. 2.7 mg of Dansyl-Cl (Sigma-Aldrich Catalog no. D2625-1G) was dissolved in 1 ml of 100% acetone to make 10 mM stock. 20 µl of polymer or L-ornithine was mixed with 20 µl of 100 mM sodium bicarbonate pH 8.3. To the dissolved peptide, 40 µl of 10 mM Dansyl-Cl was added and the mixture was incubated at 37 °C for one hour. The reaction was quenched by mixing with 20 µl of 0.1 M NaOH. Reactions with polymer were hydrolyzed by the addition of 100 µl 12 N HCl and incubation at 105 °C overnight. The next day, caps were kept open for 10 mins to dry the sample at 105 °C. The dried sample was dissolved in 50 µl of MQ. Samples were centrifuged and the supernatant was subjected to LC/MS analysis using the same column as for poly-L-ornithine analysis. HPLC method for dansylation reaction analysis consisted of a 20 min acetonitrile linear gradient from 10% to 90% in water with 0.1% formic acid at 0.5 ml/min. Mass spectrometry was performed in positive-ion mode.

**Enzymatic assay for PosA adenylation domain**. Unless otherwise stated, compounds were purchased from Sigma-Aldrich. The NADH coupled assay was performed as described by Mydy et al. [19]. To measure the adenylation activity of the truncated adenylation domain of PosA and its mutants D217A, E221A, S313A, M218A, and T304A. The reaction mixture contained 250 mM HEPES pH 7.5, 15 mM MgCl$_2$, 3 mM ATP, 150 mM hydroxylamine, 300 mM phosphoenolpyruvate, 0.8 mM NADH, 10 Units/ml of each coupling enzymes myokinase, pyruvate kinase, lactate dehydrogenase, varying concentrations of substrates (L- and D-Ornithine, L-Lysine, a stereo mixture of L/D-diamino butyrate, propionate, etc.) and 1 µM PosA adenylation domain protein. The D217A, E221A, T304A, and S313A mutants were initially examined at 1 µM concentration yielding low activity. We therefore retested with 5 to 20 µM concentration of enzyme, keeping other components at the same concentration. The assay was performed in a 100 µl reaction volume in 96-well black polystyrene plates with clear bottoms at 37 °C for 15 min. All reactions were performed in triplicate; mean and standard deviation are shown in plots. Protein and substrates were added as 10 µl aliquots in each well without contacting each other and the reaction was initiated by the addition of 80 µl Master mix containing remaining components at 1.25× final concentration. Continuous measurement was done at 340 nM absorbance using Biotek Synergy 4 plate reader at 37 °C for 15 min. Initial velocity

plots for substrate screening and kinetic plots to derive reaction constants were analyzed in GraphPad Prism 6 software. Substrate screening was performed with the same assay, performed in triplicate using 5 mM for each substrate and the same protein sample (Supplementary Fig. 3).

**Generation of posA gene knockouts in *A. baumannii***. A kanamycin resistance gene was introduced into *pos*A gene along with frame shift mutation at 527 bp. Primers AB_kan_F and AB_kan_R (Supplementary Table 3) were used to amplify kanamycin gene cassette. PCR amplification product was gel purified and transformed into competent *A. baumannii* 307-0294 strain cells using electroporation. After incubation for four hours, cells were plated on LB agar plates with 50 ug/ml kanamycin and incubated overnight. Transformant colonies were grown overnight in fresh LB medium containing 50 ug/ml kanamycin followed by genomic DNA isolation and PCR with sequencing primers AB_KO_seqF and AB_KO_seqR. The amplification product was sequenced to confirm the presence of kanamycin resistance gene cassette within *pos*A gene.

**Computational tools**

*Sequence similarity network (SSN) using EFI-EST*. An initial attempt to generate SSN using BLAST mode of EFI-EST did not include any sequences from eukaryotic phyla and a decrease in stringency led to the inclusion of other types of NRPS-like proteins. Hence, various homologous sequences of CHPAs were collected using BLAST and HMMER search in all bacterial and eukaryotic phyla individually and confirming the presence of NRPS-terminal domain. A list of all combined sequences was uploaded in FASTA mode in EFI-EST (https://efi.igb.illinois.edu/efi-est/#)[45,46]. All-by-all BLAST was performed by EFI-EST to obtain the similarities between sequence pairs and edge value calculation for SSN. The network file was generated with a sequence length cutoff 1100 to 1800 and an alignment score threshold of 525. Cytoscape was used to view, analyze, and generate images for network analysis.

*Transmembrane helix prediction*. Transmembrane helices were predicted using CCTOP[47] and DeepTMHMM (https://dtu.biolib.com/DeepTMHMM).

*Generation of phylogenetic tree*. Clustal omega[48] (http://www.ebi.ac.uk/Tools/msa/clustalo) was used to align selected sequences. The resulting phylogenetic tree was downloaded and visualized in itol (https://itol.embl.de)[49].

**Biofilm assay**. *Acinetobacter baumannii* biofilm assay was performed as described by O'Toole et al. [50]. 10 µl of overnight grown cultures diluted to OD600 of 0.1 were inoculated into 90 µl of L.B. medium in sterile 96-well plates and grown at 37 °C for 48 h without shaking. Both wild type and *pos*A knockout *Acinetobacter baumannii* 307 were grown in triplicate wells. OD600 was measured from cultures after 48 hours and considered static. For biofilm measurements, planktonic cells were decanted and biofilm on the surface of wells was washed three times with 150 µl of sterile 154 mM NaCl solution, air dried for 30 min, and stained with 150 µl of 1 % crystal violet solution for 15 min. Stain solution was decanted and wells were washed again with 150 µl of sterile 154 mM NaCl solution and air dried. Biofilm associated dye was eluted by adding 150 ul of 95% ethanol for 10 min followed by absorbance at 595 nm.

**Antimicrobial assay**. Antimicrobial assays were performed to obtain minimum inhibitory concentration for bacterial and

fungal strains. YPD medium was used for *Saccharomyces cerevisiae* BY4742, *Cryptococcus neoformans*, and *Candida albicans*, and LB medium for *Pseudomonas aeruginosa* PA01 and *Klebsiella pneumoniae*. The antimicrobial assay was done in 5 ml culture tubes in triplicates. Overnight grown cultures were added to fresh media to adjust the $OD_{600}$ at 0.1 in 150 µl media volume. Chemically synthesized 12-δPO was dissolved in sterile water and added to cultures at various concentrations in triplicate. Bacterial cultures were incubated at 37 °C and fungi cultures at 30 °C. $OD_{600}$ was monitored for up to 24 h. The lowest concentration which inhibited the growth of microbial strains compared to control was considered Minimum Inhibitory Concentration (MIC).

**Toxicity assay in cancer cells**. The toxicity of 12-δPO to cancer cells was examined in OVCAR-3 cell lines. Cells were grown overnight in RPMI media in a canted neck cell culture flask. Adherent cells were suspended in culture by trypsin treatment followed by the inactivation of trypsin by serum. Cells were dispensed in triplicates at $1 \times 10^4$ cells in 100 µl per well in 96-well cell culture plate and incubated overnight at 30 °C in $CO_2$ incubator. The following day, 30 % confluency was observed and all wells were exchanged with fresh RPMI media. Stocks of 12-δPO dissolved in 10 mM HEPES pH 7.2 were added to wells as 5 µl aliquot to achieve 100 µM, 250 µM and 500 µM concentrations in cell culture. 10 mM HEPES pH 7.2 was added as vehicle control. After two days of 12-δPO treatment, media was exchanged with trypsin solution. RPMI media with serum was added to inactivate trypsin. 10 µl of cell culture was mixed with an equal volume of trypan blue dye and cells were counted on the Automatic cell counter.

**Chemical synthesis of poly-L-ornithine standards**. Custom peptide synthesis of 12-mer δ-amino bonded δ-poly-L-ornithine (12-δPO) was performed by LifeTein peptide synthesis company (Somerset, NJ). Custom synthesized 12-mer of α-poly-L-ornithine was obtained from GenScript (Piscataway, NJ). Both peptides were provided as HPLC-purified powders.

**Statistics and reproducibility**. Apparent kinetic constants were determined with six to nine substrate concentrations and triplicate measurements of a single protein sample. Substrate specificity values were determined with triplicate measurements from a single protein sample. The biofilm measurements were performed in triplicate and examined with a paired *T*-test.

**Reporting summary**. Further information on research design is available in the Nature Portfolio Reporting Summary linked to this article.

## Data availability

The structures of the PosA adenylation domain have been deposited with the Protein Data Bank (unliganded PosA, PDB **8G95**; PosA bound to L-ornithine, **8G96**; PosA bound to D-ornithine, **8G97**; and PosA bound to L-lysine, **8G98**). Source data are provided in an enclosed file. The source data behind the graphs in Fig. 1C, D, and Supplementary figures. 3, 16, and 19 are available in Supplementary Data 1.

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

## Acknowledgements

The investigation was supported by grants to Andrew M. Gulick from NIH GM136235. Diffraction data were collected at APS (Advanced Photon Source, Lemont, IL 60439, USA) and SSRL (Stanford Synchrotron Radiation Lightsource, Menlo Park, CA 94025, USA). GM/CA@APS has been funded by the National Cancer Institute (ACB-12002) and the National Institute of General Medical Sciences (AGM-12006, P30GM138396). This research used resources of the Advanced Photon Source, a U.S. Department of Energy (DOE) Office of Science User Facility operated for the DOE Office of Science by Argonne National Laboratory under Contract No. DE-AC02-06CH11357. The Eiger 16 M detector at GM/CA-XSD was funded by NIH grant S10 OD012289. Use of the Stanford Synchrotron Radiation Lightsource, SLAC National Accelerator Laboratory, is supported by the U.S. Department of Energy, Office of Science, Office of Basic Energy Sciences under Contract No. DE-AC02-76SF00515. The SSRL Structural Molecular Biology Program is supported by the DOE Office of Biological and Environmental Research, and by the National Institutes of Health, National Institute of General Medical Sciences (P30GM133894). We like to acknowledge Uli MacDonald and Thomas Russo for help with generation of knockout mutant of *A. baumannii* 307; Murat C. Kalem, John C. Panepinto, Nicholas D. Clark, and Michael G. Malkowski for providing fungal strains; Dale Kreitler and Timothy Wencewicz for helpful discussions with chemical analysis; and Saveg Yadav and Arthur M. Edelman for help with toxicity assay in cancer cell lines.

## Author contributions

The experimental work described in the manuscript was performed by K.D.P under the supervision of A.M.G. Data analysis and interpretation were performed by K.D.P. and A.M.G. Both authors wrote, edited, and revised the manuscript.

## Competing interests

The authors declare no competing interests.
