## [Peer Review File · Communications Biology]

Reviewers' comments:

Reviewer #1 (Remarks to the Author):

Several cationic homopolyamino acids have been identified from actinomycete bacteria. They are produced by NRPS-like enzymes, which polymerize a diaminoacid monomer unit between the carboxylate and the side chain amine. The N-terminal adenylation domain (A domain) of the enzyme selects and activates the diaminoacid substrate, and the resulting aminoacyl-adenylates are polymerized by the C-terminal region. In this manuscript, the authors report the identification of a homologue enzyme (PosA) from *Acinetobacter baumannii*. They further suggested that this enzyme product was a new cationic homopolyamino acid, poly-L- δ -ornithine.

In a series of experiments, the crystal structure of the PosA-A domain was solved to understand the substrate (L-ornithine) binding, and the A domain activity was detected for L-ornithine. Although the full-length recombinant PosA was successfully expressed in *E. coli*, a trace amount of poly-L-ornithine was detected in the enzyme reaction in vitro. Therefore, its isopeptide linkage of the polymer product was not experimentally proved. In addition, the poly-L-ornithine production was suggested only with the MS spectra corresponding to 11-mer and 12-mer. This reviewer can agree about the difficulty due to the poor enzymatic activity. Still, more careful experiments using the recombinant enzyme are needed to obtain persuasive data for the enzyme product. At least, the authors need to show the LCMS data (chromatograms and MS spectra) of the reaction mixtures with and without the recombinant enzyme. The chemically synthesized poly-L- δ -ornithine should be a good reference compound to detect the polymer product in the enzymatic reaction. If the enzyme product is a new cationic amino acid polymer, knowing the isopeptide linkage of the cationic homopolyamino acid is very important.

Overall, the authors should write figure legends more accurately.

Minor clarifications/edits are suggested to improve the manuscript as follows:

Page 8, line 5

10-mer to 12-mer? Figure 2 and Supplemental Figure 6 show the data for 11-mer and 12-mer.

Page 8, lines 9-7 from the bottom.

The data for poly-L-lysine should be shown. "The data not shown" is not allowed.

Page 26, Ref.3

Ref.3 and Ref.18 must be identical. You need to check the references.

Supplemental Figures 5 and 6.

This reviewer cannot understand the contents of these Supplemental Figures because of the poor explanation in the figure legends.

Figure 3, Supplemental Figures 7, and 8

Why don't you show the critical amino acid residues in the images to explain the interaction between the enzyme and the ligand?

Supplemental Figure 9

Some data overlap with Figure 1C.

Reviewer #2 (Remarks to the Author):

This manuscript describes the functional and structural analysis of ϵ Poly-Lysine (ϵ PL) synthetase homolog PosA in *Acinetobacter baumannii*. The authors showed that the adenylation domain of PosA prefers L-ornithine as a substrate. The full-length protein of PosA produces poly-L-ornithine, which is different from homopolymers produced by the previously reported *Streptomyces* ϵ PL synthetase homologs. The authors also determined the structure of PosA adenylation domain with several diamino acid substrates, providing detailed insights into substrate recognition. Furthermore, the authors identified homologs with different substrate-binding residues based on structurally guided genome mining and phylogenetic analysis. Because this study suggests the presence of novel cationic homopolyamino acid peptide synthetases, the obtained information would be valuable for the community. The manuscript is well written. Therefore, I recommend this paper for publication after addressing the following minor issues.

1. The residue number should be added in Fig. 3, Fig. S7, and Fig. S8. Otherwise, it would be difficult to understand which residues are involved in the substrate recognition.
2. page 10: It is described that Thr304 and Ser313 are placed at hydrogen bonding distances of 3.1 and 3.0 Å from the terminal amine group of L-Orn. However, these distances are not consistent with those shown in Fig. 3B (L-Ornithine substrate molecule interaction with substrate binding residues).
3. The authors mentioned that Met218 is important for the L-ornithine preference. The M218A mutation resulted in increased activation for L-lysine and slightly decreased activation for L-ornithine. However, the M218A mutant still showed higher activity against L-ornithine compared with L-lysine. Are there any other important residues for the L-ornithine preference? For example, Ser313 of PosA is replaced with Val in ϵ PL synthetase (7WEW). Considering that the S313A mutant showed a significantly reduced catalytic efficiency, Ser313 might also be an important residue for the L-ornithine preference.

We provide below a point-by-point response to the reviews, and additionally provide a new manuscript that highlights the changes that we have made in response to their concerns. We thank you and both reviewers for their careful reviews, which has resulted in an improved manuscript. In the main text of the revised manuscript, the new material highlighted in red font. Minor changes and edits that merely improve the clarity are not highlighted.

Thank you.

Ketan D. Patel and Andrew M. Gulick
University at Buffalo

Reviewers' comments:

Reviewer #1 (Remarks to the Author):

Several cationic homopolyamino acids have been identified from actinomycete bacteria. They are produced by NRPS-like enzymes, which polymerize a diaminoacid monomer unit between the carboxylate and the side chain amine. The N-terminal adenylation domain (A domain) of the enzyme selects and activates the diaminoacid substrate, and the resulting aminoacyl-adenylates are polymerized by the C-terminal region. In this manuscript, the authors report the identification of a homologue enzyme (PosA) from *Acinetobacter baumannii*. They further suggested that this enzyme product was a new cationic homopolyamino acid, poly-L- δ -ornithine.

In a series of experiments, the crystal structure of the PosA-A domain was solved to understand the substrate (L-ornithine) binding, and the A domain activity was detected for L-ornithine. Although the full-length recombinant PosA was successfully expressed in *E. coli*, a trace amount of poly-L-ornithine was detected in the enzyme reaction in vitro. Therefore, its isopeptide linkage of the polymer product was not experimentally proved. In addition, the poly-L-ornithine production was suggested only with the MS spectra corresponding to 11-mer and 12-mer. This reviewer can agree about the difficulty due to the poor enzymatic activity. Still, more careful experiments using the recombinant enzyme are needed to obtain persuasive data for the enzyme product. At least, the authors need to show the LCMS data (chromatograms and MS spectra) of the reaction mixtures with and without the recombinant enzyme. The chemically synthesized poly-L- δ -ornithine should be a good reference compound to detect the polymer product in the enzymatic reaction. If the enzyme product is a new cationic amino acid polymer, knowing the isopeptide linkage of the cationic homopolyamino acid is very important.

We would like to thank reviewer for the effort and time in reviewing this article and providing useful comments. New experiments were performed to include all controls asked by reviewer (Figure 2 in revised manuscript). We have improved the purification of the enzyme using n-dodecyl- β -D-maltopyranoside (DDM), which has allowed us to repeat the biochemical experiment. The PosA enzymatic reaction described in Materials and Methods was performed with L-Ornithine and L-Lysine, as well as “no protein” control. Enzymatic reactions were passed over SP cation exchange and samples from elution fractions were analyzed using LCMS (Figure 2A). LCMS chromatograms without protein, L-Lysine reaction and Poly- δ -ornithine standard

are updated in Figure 2B and 2D. MS spectra representing 7-mer to 12-mer of Poly- δ -ornithine from PosA reaction were observed and are updated in figure 2C.

Additionally, we performed a series of experiments to demonstrate that the polymers are linked through isopeptide bonds with the side chain. Our original manuscript employed 12-mers of δ -poly-L-ornithine as a standard. For the current experiments, we additionally purchased a 12-mer of α -poly-L-ornithine. With these two standards, we were motivated by the chemical labeling approach previously employed by Yamanaka in the characterization of the lysine and β -lysine polymers, although we used a different chemical labeling agent (Supplementary Figure 10).

We developed a two-step protocol to address the nature of the PosA product. In four separate reactions, free ornithine, the two standards (poly- δ and poly- α), as well as the PosA poly-ornithine enzymatic product, were chemically labeled with dansyl chloride, which reacts with the free amines. We then hydrolyzed the peptides producing singly labeled ornithine monomers. Dansylation of free ornithine was done with a 1:1 ratio of dansyl chloride to ornithine, to produce mostly singly labeled monomers. The labeled ornithine monomers were run on a LC/MS and monitored by OD₂₅₄ (for the dansyl group) and mass spectrometry (Figure 3). Three peaks that absorbed at 254 nm were observed and identified as a small peak of dansyl hydroxide and two peaks of a mass that was consistent with the α - or δ -labeled ornithine. We used the two polymer standards to identify the nature of the two dansyl ornithine peaks. Due to the availability of the free amines, dansylation and hydrolysis of δ -poly-L-ornithine results in the formation of α -dansyl ornithine, while the protocol with α -poly-L-ornithine results in δ -dansyl ornithine. LC/MS with the two polymer standards results exclusively in one of the two dansyl ornithine peaks, clearly identifying the identity (Figure 3). This work is supported by Supplementary Figures 11-13 and Supplementary Table 1, in the revised manuscript.

The products of the PosA reaction were then subjected to dansylation and hydrolysis, resulting exclusively in a peak that comigrates with α -dansyl ornithine. The experiment demonstrates unequivocally that the PosA polymer product is δ -poly-L-ornithine, joined via the isopeptide bonds between the side chain amine and the carboxylate. These data are now shown in Figure 3 in revised manuscript.

Overall, the authors should write figure legends more accurately.

Thank you. We have reviewed all of the figure legends, and provide clearer descriptions throughout.

Minor clarifications/edits are suggested to improve the manuscript as follows:

Page 8, line 5

10-mer to 12-mer? Figure 2 and Supplemental Figure 6 show the data for 11-mer and 12-mer.

Thank you for pointing the missing data for 10-mer. Indeed, LCMS analysis of the concentrated the poly-L-ornithine from PosA reaction showed the presence of 7-mer to 12-mer. MS spectra for concentrated sample is shown in Figure 2C. EIC data for the alternate polymer lengths are shown in Supplementary Figure 8.

Page 8, lines 9-7 from the bottom.

The data for poly-L-lysine should be shown. "The data not shown" is not allowed.

These data are now included in Figure 2 and Supplementary Figures 6-8 using new experiments with DDM purified PosA full length protein.

Page 26, Ref.3

Ref.3 and Ref.18 must be identical. You need to check the references.

Thank you for your careful review; we have corrected the references.

Supplemental Figures 5 and 6.

This reviewer cannot understand the contents of these Supplemental Figures because of the poor explanation in the figure legends.

The figure legend has been completely re-written for SI Figure 5. References for the software employed to predict the presence of transmembrane helices is provided. SI Figure 6 has been replaced by several new figures. We were careful to describe the MS figures in detail to allow the reader to understand our analysis of the polymer.

Figure 3, Supplemental Figures 7, and 8

Why don't you show the critical amino acid residues in the images to explain the interaction between the enzyme and the ligand?

The critical amino acid residues having the interactions with ligand are now shown with labels. The new organization of the material has these in Supplementary Figures 14 and 15.

Supplemental Figure 9

Some data overlap with Figure 1C.

The Michaelis-Menten plots for WT are repeated in panel A of the Supplementary Figure 9 (now Supplementary Figure 16) to facilitate comparison with the plots of mutants. We respectfully prefer to keep these panels in both places.

Reviewer #2 (Remarks to the Author):

This manuscript describes the functional and structural analysis of ϵ Poly-Lysine (ϵ PL) synthetase homolog PosA in *Acinetobacter baumannii*. The authors showed that the adenylation domain of PosA prefers L-ornithine as a substrate. The full-length protein of PosA produces poly-L-ornithine, which is different from homopolymers produced by the previously reported *Streptomyces* ϵ PL synthetase homologs. The authors also determined the structure of PosA adenylation domain with several diamino acid substrates, providing detailed insights into substrate recognition. Furthermore, the authors identified homologs with different substrate-binding residues based on structurally guided genome mining and phylogenetic analysis. Because this study suggests the presence of novel cationic homopolyamino acid peptide

synthetases, the obtained information would be valuable for the community. The manuscript is well written. Therefore, I recommend this paper for publication after addressing the following minor issues.

We are thankful to the reviewer for their effort and time and the useful comments and helpful suggestions. We have revised the manuscript as per their suggestions.

1. The residue number should be added in Fig. 3, Fig. S7, and Fig. S8. Otherwise, it would be difficult to understand which residues are involved in the substrate recognition.

Thank you for the suggestion, the amino acid residues are labelled in all the figures (Figures 4 and Supplementary Figures 14 and 15)

2. page 10: It is described that Thr304 and Ser313 are placed at hydrogen bonding distances of 3.1 and 3.0 Å from the terminal amine group of L-Orn. However, these distances are not consistent with those shown in Fig. 3B (L-Ornithine substrate molecule interaction with substrate binding residues).

Thank you. The correct distances are updated in the manuscript.

3. The authors mentioned that Met218 is important for the L-ornithine preference. The M218A mutation resulted in increased activation for L-lysine and slightly decreased activation for L-ornithine. However, the M218A mutant still showed higher activity against L-ornithine compared with L-lysine. Are there any other important residues for the L-ornithine preference? For example, Ser313 of PosA is replaced with Val in ϵ PL synthetase (7WEW). Considering that the S313A mutant showed a significantly reduced catalytic efficiency, Ser313 might also be an important residue for the L-ornithine preference.

Ser313 and Thr304 do appear, along with Met218, to contribute to the preference for L-Ornithine. As shown in Supplemental Table 2 for known CHPAS, Ser is also present in Stachelhaus code for D-DAB substrate and unknown substrate in *Sulfuriferula plumbiphila* (WP_147073502.1). However, Methionine is only present in L-Ornithine activating CHPAS. Furthermore, there appears to be a wider diversity of residues at position 2, suggesting this could be more important residue comparatively for L-Ornithine preference.

REVIEWERS' COMMENTS:

Reviewer #1 (Remarks to the Author):

The revised manuscript by Patel et al has been adjusted based on the reviewer's comments. The authors have adequately answered all the questions raised by the reviewers and have improved their manuscript, which is now acceptable for publication by communications biology.